# Home interventions and light therapy for the treatment of vitiligo (HI-Light Vitiligo Trial): study protocol for a randomised controlled trial

Rachel H Haines,[1] Kim S Thomas,[2] Alan A Montgomery,[1] Jane C Ravenscroft,[3] Perways Akram,[4] Joanne R Chalmers,[2] Diane Whitham,[5] Lelia Duley,[5] Viktoria Eleftheriadou,[2] Garry Meakin,[1] Eleanor J Mitchell,[1] Jennifer White,[1] Andy Rogers,[4] Tracey Sach,[6] Miriam Santer,[7] Wei Tan,[1] Trish Hepburn,[1] Hywel C Williams,[2] Jonathan Batchelor[2]

For numbered affiliations see end of article.

**Correspondence to**
Professor Kim S Thomas;
kim.thomas@nottingham.ac.uk

## ABSTRACT

**Introduction** Vitiligo is a condition resulting in white patches on the skin. People with vitiligo can suffer from low self-esteem, psychological disturbance and diminished quality of life. Vitiligo is often poorly managed, partly due to lack of high-quality evidence to inform clinical care. We describe here a large, independent, randomised controlled trial (RCT) assessing the comparative effectiveness of potent topical corticosteroid, home-based hand-held narrowband ultraviolet B-light (NB-UVB) or combination of the two, for the management of vitiligo.

**Methods and analysis** The HI-Light Vitiligo Trial is a multicentre, three-arm, parallel group, pragmatic, placebo-controlled RCT. 516 adults and children with actively spreading, but limited, vitiligo are randomised (1:1:1) to one of three groups: mometasone furoate 0.1% ointment plus dummy NB-UVB light, vehicle ointment plus NB-UVB light or mometasone furoate 0.1% ointment plus NB-UVB light. Treatment of up to three patches of vitiligo is continued for up to 9 months with clinic visits at baseline, 3, 6 and 9 months and four post-treatment questionnaires. The HI-Light Vitiligo Trial assesses outcomes included in the vitiligo core outcome set and places emphasis on participants' views of treatment success. The primary outcome is proportion of participants achieving treatment success (patient-rated Vitiligo Noticeability Scale) for a target patch of vitiligo at 9 months with further independent blinded assessment using digital images of the target lesion before and after treatment. Secondary outcomes include time to onset of treatment response, treatment success by body region, percentage repigmentation, quality of life, time-burden of treatment, maintenance of response, safety and within-trial cost-effectiveness.

**Ethics and dissemination** Approvals were granted by East Midlands—Derby Research Ethics Committee (14/EM/1173) and the MHRA (EudraCT 2014-003473-42). The trial was registered 8 January 2015 ISRCTN (17160087). Results will be published in full as open access in the NIHR Journal library and elsewhere.

**Trial registration number** ISRCTN17160087.

## Strengths and limitations of this study

► Answers research questions prioritised by people with vitiligo and healthcare professionals as a part of a James Lind Alliance Priority Setting Partnership.
► Appropriately powered, pragmatic, randomised controlled trial, which may influence clinical decision making for a condition with little high-quality evidence to support treatment options.
► Trial places emphasis on patient-reported outcomes and includes the core outcome set for vitiligo.
► Blinding of treatment allocation may be compromised due to observable side effects (eg, erythema and tanning of the surrounding skin), particularly for those receiving active light therapy.
► Primary outcome is based on participant-reported treatment success at 9 months (Vitiligo Noticeability Scale), further improvement may occur after this time as residual hyperpigmentation resolves.

## INTRODUCTION

Vitiligo is a chronic progressive condition causing loss of skin pigmentation. It affects around 0.5%–1% of the world's population, and can develop at any age, although onset between the age of 10 and 30 years is most common.[1–4] Vitiligo can have a major impact on the quality of life of affected people,[5] who often experience psychological problems such as shame, depression and anxiety, low self-esteem and social isolation.[6 7]

Current clinical guidelines for the management of vitiligo recommend narrowband ultraviolet-B light (NB-UVB), topical corticosteroids, topical tacrolimus and combination therapies.[8 9] However, the evidence base for treatments is currently limited.[10 11]

A Cochrane systematic review assessing interventions for vitiligo was updated in

2015.[12] This review identified 96 randomised controlled trials (RCTs) of vitiligo treatments, involving 4512 participants. Despite this large number of trials, the quality of the included studies was generally poor, making it difficult to make firm recommendations for vitiligo treatment. The majority of the studies included fewer than 50 participants, were at high or unclear risk of bias and few involved children. Nevertheless, the Cochrane review identified some evidence in support of NB-UVB, and the combination of NB-UVB with other active interventions (eg, topical corticosteroids or calcineurin inhibitors) appeared to be more effective than monotherapy. A 2016 review of the use of NB-UVB for the treatment of vitiligo[13] also concluded that guidance on dosing and administration of NB-UVB requires further study.

There is some evidence to suggest that starting vitiligo treatment when lesions first appear is likely to result in better treatment response, when vitiligo patches are actively changing and some melanin is still present.[14] However, current practice is to reserve NB-UVB treatment for patients with widespread and established vitiligo,[8] as the treatment requires thrice weekly visits to the hospital for full body light therapy. The recent availability of hand-held NB-UVB units for private purchase on the open market means that early intervention using treatment at home is now a possibility. Benefits of home-based treatment, if shown to be safe and effective, would be a reduction in time and financial burden for patients and hospital services, and reduced exposure of healthy skin to NB-UVB light as treatment can be targeted more effectively. Our small pilot RCT, involving 29 participants, compared hand-held NB-UVB with dummy devices for the management of vitiligo.[15] This pilot demonstrated that people with vitiligo were keen to use home-based light therapy, and that the devices were safe and well tolerated when used in a domestic setting.

A James Lind Alliance Priority Setting Partnership identified a 'top 10' list of priority areas for vitiligo research that are important to people with vitiligo and healthcare professionals responsible for their care.[16] Two of these prioritised areas relate to the use of light therapy for the management of vitiligo: 'Which treatment is more effective for vitiligo: steroid creams/ointments or light therapy?' and 'How effective is UVB therapy when combined with creams or ointments in treating vitiligo?' The prioritisation work was shared with the NIHR Health Technology Assessment Programme who subsequently invited applications to address these two priority topics in open competition. The contract to conduct the commissioning brief was awarded to this team.

## Objectives
Primary objective:
1. To evaluate the comparative safety and effectiveness of home-based interventions (potent topical corticosteroids and hand-held NB-UVB light) for the management of early and limited vitiligo in adults and children. Specifically, we are comparing:

 a. Potent topical corticosteroid (mometasone furoate 0.1% ointment) with hand-held NB-UVB light;

 b. Potent topical corticosteroid (mometasone furoate 0.1% ointment) with combination of hand-held NB-UVB light and potent topical corticosteroid.

Secondary objectives:
1. To assess whether treatment response (if any) is maintained once the intervention is stopped.
2. To compare the cost-effectiveness of the interventions from a National Health Service (NHS) and a family perspective.

## METHODS
The analysis and reporting of the trial will be in accordance with Consolidated Standards of Reporting Trials guidelines.[17]

### Study design and setting
The trial is a multicentre three-arm, parallel group, pragmatic, placebo-controlled RCT, recruiting adults and children with early and limited vitiligo. The trial treatments are home-based therapies (potent topical corticosteroid and hand-held NB-UVB), and their use is prescribed and overseen in a secondary care setting across 18 hospitals in the UK. A full list of recruiting centres can be found in the 'Acknowledgements' section. A mixed-methods process evaluation is being conducted alongside the HI-Light Vitiligo Trial. The full protocol is available at www.vitiligostudy.org.uk.

### Participants
#### Eligibility
##### Inclusion criteria
Participants aged 5 years and over with a diagnosis of non-segmental vitiligo, limited to approximately 10% or less of body surface area, and at least one vitiligo patch that has been active in the last 12 months (reported by the participant, or parent). Participants should be willing to stop any other active therapies for their vitiligo at time of randomisation, be able to administer the trial treatments safely at home (able to follow the treatment instructions and children able to comply with the necessary safety precautions). Participants also need to be willing and able to give informed consent (or parental/guardian consent in the case of children).

##### Exclusion criteria
Potential participants with segmental or universal vitiligo, vitiligo limited to areas contraindicated for treatment with potent topical corticosteroid (eg, around the genitals) or evidence of marked Koebner phenomenon (lesions appearing in sites of skin trauma) as such potential participants are likely to require urgent care. Also excluded are potential participants with a history of skin cancer, radiotherapy use or photosensitivity; women who are pregnant,

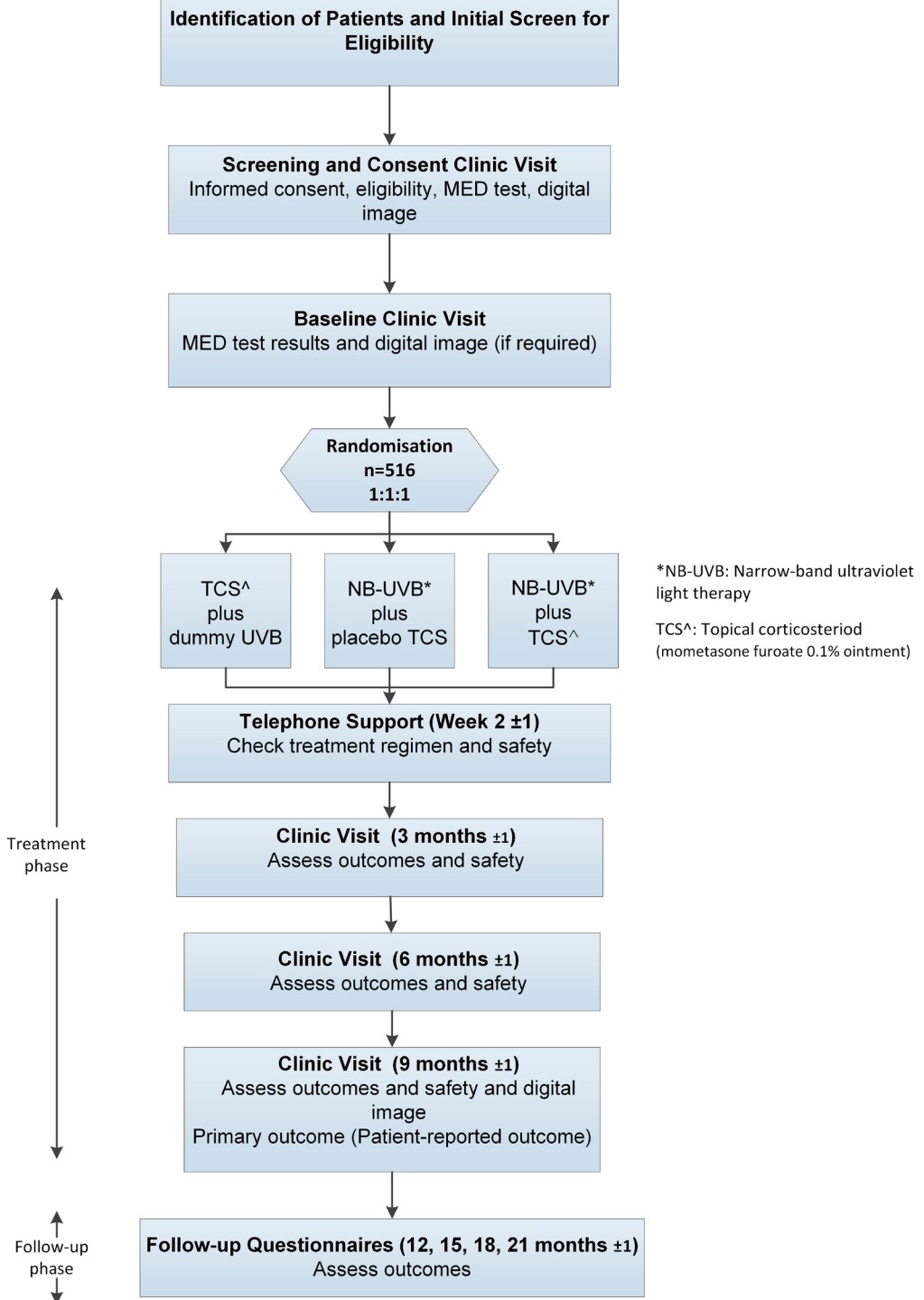

**Figure 1** Flow of participants through the trial. MED, minimum erythema dose.

breastfeeding or likely to become pregnant during the 9-month treatment period; those currently using immunosuppressive drugs, or involved in another clinical trial and those with allergy or contraindication to mometasone furoate or any of its excipients. Potential participants are not randomised into the trial if the research nurse or investigator feels that they are unable to follow the treatment instructions, or if children are unable to comply with the necessary safety precautions.

### Participant timeline

Participants are enrolled in the trial for a total of up to 21 months (9 months treatment phase, up to 12 months follow-up). During the treatment phase, participants attend a hospital clinic five times (two consecutive days at baseline for recruitment, medical photography, photosensitivity assessment (minimum erythema dose (MED) test) and training in use of the interventions), followed by assessment at 3, 6 and 9 months for compliance and outcome. The additional follow-up is by 3-monthly

questionnaires, self-completed by post or email link. Due to trial timelines, participants recruited towards the end of the recruitment may be followed-up for a shorter period of time.

Flow of participants through the trial is summarised in figure 1, and details of the data collection schedule are summarised in table 1.

## Recruitment

Potential participants are identified through secondary care dermatology and paediatric clinics, through mailshots from general practices and by self-referral as a result of community advertising and trial publicity. Potential participants are initially screened by telephone in order to ensure that only people who are potentially eligible to take part are invited to attend a clinic appointment.

## Randomisation and blinding

Participants are randomised to one of three treatment groups in a ratio of 1:1:1 as follows:

► Topical corticosteroid ointment plus dummy NB-UVB light (corticosteroid only);
► Vehicle ointment plus NB-UVB light (light therapy only);
► Topical corticosteroid ointment plus NB-UVB light (combination therapy).

Randomised allocation to treatment groups is minimised by recruiting centre, body region of target patch (face and neck, hands and feet or rest of the body) and age (5–16 years or >16 years), weighted towards minimising the imbalance in trial arms with a probability of 0.8. After all eligibility criteria have been confirmed and medical photographs taken, participants are randomised by staff at the recruiting hospital via a secure web server created and maintained by the Nottingham Clinical Trials Unit.

Participants are only randomised into the trial once they have completed the training in use of the trial interventions, the results of the MED test are known and the participant has attended medical photography for their baseline digital photograph.

Participants, research nurses, principal investigators, members of trial management group and data analysts are blinded to treatment allocation. Only the Nottingham Clinical Trials Unit (NCTU) programmer, the medical physics staff, pharmacy staff and the NCTU Quality Assurance staff have access to the treatment allocation schedule for the sole use of preparing, blinding and checking of the trial treatments.

While every effort is made to maintain blinding of the trial interventions, there is a risk that participants may be able to guess their treatment allocation as a result of side effects of the interventions (eg, redness and tanning of the skin). To minimise detection bias, outcomes will be assessed by blinded assessors (both independent clinicians and a panel of patients), using digital images taken at baseline and at 9 months. Participant information leaflets will emphasise that all participants will receive at least one active treatment for their vitiligo, thus reducing potential unblinding due to lack of treatment response. During our previous trial in which participants were randomised to active or dummy light therapy devices, 70% (19/27) of participants and 40% (16/27) of research nurses, guessed treatment allocation correctly. The main reasons for unblinding of the research nurses were redness of the skin (30% (3/10)) and improvement in vitiligo (60% (6/10)) in the active group, and lack of treatment response (100% (6/6)) in the placebo group.

## Interventions

### Topical therapy: potent topical corticosteroid

Mometasone furoate 0.1% ointment (Elocon, Merck, Sharp and Dohme, PL00025/0578), a potent corticosteroid used once daily, has been recommended in the European Clinical Guidelines for the management of vitiligo.[9] In order to minimise the risk of adverse reactions, the Guidelines recommend a discontinuous regimen involving periods of use followed by break periods. Possible adverse reactions to mometasone furoate 0.1% (as listed in the Summary of Product Characteristics[18]) include: infection, folliculitis, paraesthesia, burning sensation, contact dermatitis, skin hypopigmentation, hypertrichosis, skin striae, acneiform dermatitis, skin atrophy, pruritus, application site pain and visual disturbance. Participants are advised to stop treatment with the ointment if they notice any side effects and to contact the local research team for review and advice on when to restart treatment.

### Topical therapy: vehicle ointment

The vehicle ointment is an inert ointment (white soft paraffin) present in the base of mometasone furoate ointment. It is relatively free of side effects.[19]

### Topical therapy: treatment regimen

To minimise the risk of adverse reactions, topical therapy is to be applied as a thin layer to the affected patches once daily on alternate weeks (1 week on, 1 week off), for a period of 9 months. Participants are instructed that topical therapy should be applied at least 2 hours after the hand-held light device is used in order to avoid interaction of the NB-UVB light and topical corticosteroid.

Participants are initially supplied with two x 90 g tubes of ointment after randomisation. Additional tubes are prescribed by the investigator as required.

### Light therapy: NB-UVB device

Several brands of hand-held NB-UVB units are CE marked for use in treating vitiligo and suitable for use at home. Dermfix 1000 MX units are being used for this trial.

Known adverse reactions to NB-UVB light include: erythema, blistering, burns, pruritus, perilesional hyperpigmentation, hypersensitivity reactions, cold sores and dry skin. Potential long-term risks include skin ageing and increased risk of skin cancer. Side effects can be reduced by appropriate use of the device.

**Table 1**  Timetable of study assessments

| Outcome collected | Prescreen | Visit 1 screening | Visit 2 baseline | Telephone screen (2 weeks) | Visit 3 (3 months) | Visit 4 (6 months) | Visit 5 (9 months) | Questionnaires (12, 15, 18 months) | Questionnaire (21 months) |
|---|---|---|---|---|---|---|---|---|---|
| Eligibility checks | ✓ | ✓ | ✓ | | | | | | |
| Consent | | ✓ | | | | | | | |
| Minimum erythema dose (MED) test | | ✓ | | | | | | | |
| Baseline characteristics (alternative timing) | | ✓ | (✓) | | | | | | |
| Digital images (alternative timing) | | ✓ | (✓) | | | | ✓ | | |
| Training session (alternative timing) | | ✓ | (✓) | | | | | | |
| Supervised treatment session | | ✓ | (✓) | | | | | | |
| MED test results | | | ✓ | | | | | | |
| Randomisation | | | ✓ | | | | | | |
| Telephone support check | | | | ✓ | | | | | |
| Vitiligo Noticeability Scale | | | | | ✓ | ✓ | ✓ | ✓ | ✓ |
| % repigmentation | | | | | ✓ | ✓ | ✓ | | |
| Onset of treatment response | | | | | ✓ | ✓ | ✓ | | |
| Maintenance of treatment response | | | | | | | | ✓ | ✓ |
| Quality of life questionnaires | | ✓ | | | ✓ | ✓ | ✓ | ✓ | ✓ |
| Adverse reactions | | | | ✓ | ✓ and Diary | ✓ and Diary | ✓ and Diary | | |
| Treatment usage and adherence | | | | ✓ | ✓ and Diary | ✓ and Diary | ✓ and Diary | | |
| Health resource use | | ✓ | | | ✓ and Diary | ✓ and Diary | ✓ and Diary | ✓ | ✓ |

## Light therapy: dummy device

The dummy light therapy device is identical to the active device, with the exception that a specially designed spacer comb, identical to that found on the active device, is used to block the transmission of NB-UVB light to the skin. Active and dummy devices are tracked using manufacturer's serial numbers. Experience from our pilot trial[15] has shown that the use of a dummy device is acceptable to patients and is effective in blocking the UVB radiation.

There are no known side effects of the dummy NB-UVB devices.

## Light therapy: quality control prior to distribution

All light therapy devices are tested for safety and output by the Medical Physics Department at Nottingham University Hospital NHS Trust before sending to participants. If an active device is found to have an output that is ±10% of the expected mean output, or if a dummy device tests positive for any NB-UVB emission, the device is returned to the manufacturer. Any device which is damaged or ceases to function during the treatment phase is replaced with a new unit.

## Light therapy: regimen

Although NB-UVB (UV radiation wavelengths of 311–312 nanometres) is now the most common form of light therapy used to treat skin conditions, many gaps remain in knowledge about its use. In a 2016 paper,[13] Madigan *et al* published a list of 12 key questions regarding the use of narrowband UVB for generalised vitiligo. How each of these questions has been addressed within the context of the HI-Light trial is presented in table 2. Prior to randomisation, all participants receive an MED test, to ensure eligibility for the trial. Results of the MED test are not used to determine starting dose of the light therapy, but instead to ensure that the participant does not have any undiagnosed photosensitivity disorder. All participants follow a predefined treatment schedule for the light treatment, with a starting dose of $0.05 \, \text{J/cm}^2$ (see supplementary appendix 1).

Treatment is self-administered at home, every other day, for a period of 9 months. If the vitiligo is on the face, or in an area that is inaccessible, participants are advised to seek help from someone else to administer the treatment. Parents/guardians are instructed in how to administer the treatment for their child.

The exposure time for each session is increased incrementally. Participants are asked to keep a record of their treatments in a treatment diary. Details are provided to participants on how to progress through the treatment schedule, step down or temporarily stop, depending on treatment response (a summary of instructions can be found in online supplementary appendix 2). Full instructions from the participants' handbook can be found at www.vitiligostudy.org.uk. All devices are supplied with gloves and UV protective goggles (for both the participants and their helper as required), which must be worn at all times during use.

## Storage and distribution of trial treatments

A central pharmacy and distribution centre (Mawdsleys, Doncaster, UK) receives blinded interventions. On randomisation of a participant by the trial investigator/nurse, the pharmacy is notified of the container numbers of ointment and the device to be allocated to that participant via a web-based system. Trial treatments are then sent directly to the participant's home following check and release by a qualified person.

## Training in use of interventions

Before randomisation, all participants receive training in how to apply the ointment, including guidance on avoiding application to the eyelids and sensitive body sites such as the genital area. In addition, participants receive training in the correct use of the light therapy devices. A specially developed online training video is available (www.vitiligostudy.org.uk) and face-to-face training with the research nurse or investigator is given, covering how to record treatments, manage side effects and use of the treatment schedule. Written instructions are also provided for the participants to take home. Training takes place prior to randomisation and participants have the opportunity to ask questions and points of clarification. Anyone considered unable to follow the treatment regimen is excluded from the trial. Two weeks after randomisation, participants are telephoned by the trial team to check how they are getting on with the interventions and to confirm their understanding of treatment usage and completion of the treatment diaries. Additional training on use of either treatment is provided to the participants at this time point (over the telephone or face-to-face), if needed.

## Choice of vitiligo patches for treatment

During the baseline clinic appointment, participants are asked to select up to three patches of vitiligo for treatment; one for each of three anatomical regions (head and neck, hands and feet and rest of body). One of these three patches is selected by the participants as being the patch that they would most like to see an improvement in. This will be used as the target patch for the primary outcome assessment. Vitiligo is known to respond differently at different body sites, with the face and neck being more likely to respond to treatment than the hands and feet.[20] Training material provided to sites advised nurses to inform participants that patches on the hands and feet may be more difficult to treat so they may wish to choose a target patch from one of the other body regions.

If participants wish to treat additional patches of vitiligo they are free to do so, but the additional patches are not assessed as part of the trial. Participants are encouraged to consider the additional time burden of treating more than three patches, and to prioritise treatment of the study patches if the time burden becomes restrictive.

| | Table 2 | Key questions regarding the use of NB-UVB for generalised vitiligo[13] |
|---|---|---|
| | **Question** | **Strategy tested in the HI-Light Vitiligo Trial** |
| 1 | What is the optimal weekly frequency of NB-UVB treatment? | HI-Light Trial: every other day (3–4 times weekly). Rationale: this is the most commonly used treatment regimen in the UK. |
| 2 | With regard to initial dosing, which strategy should ideally be employed? | HI-Light Trial: all participants started on the same low dose, $0.05\,J/cm^2$. Rationale: MED test was carried out before treatment, but only to identify any undiagnosed cases of photosensitivity. Starting at a fixed low dose to minimise the risk of symptomatic erythema was felt to be safer for home delivery of NB-UVB. |
| 3 | At subsequent treatments, what increments should be used for dose escalation in the absence of perceptible erythema? | HI-Light Trial: 10% dosing increase after each treatment not followed by erythema. Rationale: this reflects typical clinical practice in UK phototherapy services |
| 4 | What is the maximum acceptable dose to be given in a single treatment? | HI-Light Trial: maximum dose in the trial is $2.81\,J/cm^2$. Rationale: this reflects typical clinical practice in UK phototherapy services. |
| 5 | What is the ideal practice for dose adjustment following symptomatic erythema? | HI-Light Trial: patient self-adjustment for grades 1 and 2 erythema (according to flow chart in patient handbook) and investigator adjusted dosing for grades 3 and 4. Rationale: the upwards and downwards dosing used in the trial reflects the clinical practice of most UK phototherapy services. |
| 6 | How should the protocol be adjusted for missed doses? | HI-Light Trial: varies in function of number of missed treatments. 1 or 2 missed: go back one step on treatment schedule; 3 missed: go back two steps on treatment schedule; 4–6 missed: 50% of last dose; 6+ missed restart treatment schedule from beginning. Rationale: this conservative approach ensures that participants who have missed a lot of doses are not at risk of symptomatic erythema when they restart treatment. |
| 7 | How should a 'course' of NB-UVB therapy be defined? (ie, At what interval should further exposure be reassessed?) | *Not directly applicable within the scope of the trial.* |
| 8 | What is the maximum number of exposures allowable for patients with vitiligo given the potential risk of carcinogenesis with NB-UVB? | *Not directly applicable within the scope of the trial.* Participants in the trial will only be treating limited areas of skin and the total number of treatments will be less than the current maximum recommended number of treatments. |
| 9 | Should dosing strategies differ when treating children with vitiligo? | HI-Light Trial: children are treated in the same way as adults. Parents are given the choice of what patches they are comfortable treating, and may opt out of treating sensitive areas if they wish to do so. Rationale: the home-based treatment is more flexible than hospital-based full-body treatment, so it is possible for children to be treated in the same way as adults. |
| 10 | Should shielding of sensitive structures (eyelids, areolas and genitals) be a universal requirement, or is it safe to expose these areas if affected by vitiligo? | HI-Light Trial: the trial excludes treatment of vitiligo in the genital region. Other sensitive areas can be treated if they are affected by vitiligo, but will not otherwise be exposed to NB-UVB due to the localised nature of treatment using a hand-held device. If treating the eyes, patients are advised to seek assistance from someone else so that they can keep their eyes closed during treatment, thus reducing the risk of accidental exposure during treatment. |
| 11 | What is the most accurate definition of treatment unresponsiveness? | HI-Light Trial: treatment response is assessed in terms of its onset; unresponsiveness would be defined by patient report of 'stayed the same' or 'got worse' in response to the question, 'Compared with the start of the study, has there been a change in the vitiligo patch?' Participants are encouraged to continue treatment for as long as they are happy to do so. The trial may provide useful data on when and how to define treatment unresponsiveness. |
| 12 | How frequently should patients with vitiligo undergo surveillance following completion of a NB-UVB treatment protocol for both signs of relapse and adverse events? Is there a role for phototherapy in maintenance following repigmentation? | HI-Light Trial: long-term treatment response is being assessed 3-monthly for 1 year following completion of NB-UVB treatment. The trial has not been designed to evaluate the use of intermittent treatment for maintenance of response. Rationale: patients are particularly interested in how long treatment response might last and this is now a core outcome domain for vitiligo clinical trials. |

MED, minimum erythema dose; NB-UVB, narrowband ultraviolet B-light

## Adherence

Adherence is recorded in a treatment diary, which acts as an aide memoire to inform clinic appointment questions about the number of light treatment sessions and ointment applications undertaken since the last clinic visit. Collection of these data at each clinic visit (every 3 months)

also helps the nurse to check participants' understanding of treatment and to encourage adherence; additional training is provided at the clinic visit if it has been noticed the participant is not using the device appropriately or has misunderstood the treatment schedule.

### Concomitant medications

Some medications can occasionally cause photosensitivity (a rash in areas of skin exposed to light), although the risk of these reactions is low for NB-UVB light. As such, no changes to existing medications are required at the onset of the trial.

At the end of the 9-month treatment phase, participants return their device and any unused ointment to the trial team and are asked not to use any active treatments for their vitiligo during the follow-up period (if possible), so that the duration of any observed treatment effect can be evaluated. If participants do start active vitiligo therapy, this is recorded in the 3-monthly questionnaires.

### Treatment modification following adverse events

Participants are instructed to record adverse events in their treatment diaries and to contact their recruiting centre if they experience side effects of concern, or a serious adverse event (whether related to the trial treatment or not). For treatment-related side effects, or drug-induced photosensitivity, the site research team provide telephone advice or arrange for a dermatology consultation, as necessary. Treatment modifications, including reduction or suspension (temporary or permanent) of either topical corticosteroid or light therapy use is at the discretion of the trial nurse or dermatologist, as appropriate.

In case of a medical emergency where an active treatment of the ointment or the device would need to be stopped, active treatment of both interventions should be assumed. If knowledge of a participant's allocation is necessary, the local investigator can access an online blind-break system held by NCTU, which is available 24 hours a day.

### Outcomes

Previous research has demonstrated a discrepancy between what is collected in vitiligo trials and the outcomes that patients feel are most important to them.[21] An international e-Delphi consensus exercise[22] established core outcome domains for vitiligo trials, including: repigmentation, cosmetic acceptability of treatment response, maintenance of gained repigmentation, cessation of spread, quality of life, burden of treatment and safety. The HI-Light Vitiligo Trial will assess all of these core outcome domains.

### Primary outcome

The primary outcome of treatment success will be assessed at 9 months for a target patch of vitiligo for each participant: defined as the patch that the participant would most like to see an improvement in. Treatment success is defined as a participant's report that their vitiligo is either 'a lot less noticeable' or 'no longer noticeable' in response to the question: 'Compared with the start of the study, how noticeable is the vitiligo now?' using the validated Vitiligo Noticeability Scale (VNS).[23] Participants are shown a digital image of their vitiligo at baseline to inform this judgement at 9 months. Preliminary validation of the VNS has shown it to have good face validity with patients and good construct validity, acceptability and interpretability.[23]

Digital images of the target patch (taken at baseline and 9 months) will be assessed by a panel of independent assessors comprising three patients with vitiligo, thus providing blinded assessment of the primary outcome to explore the impact of detection bias, due to potential unblinding of the trial participants, in sensitivity analysis.

### Secondary outcomes during treatment phase

a. *Onset of treatment response:* assessed by participant and investigator for each patch of vitiligo in each of the three body regions. The question 'Compared with the start of the study, has there been a change in the vitiligo patch?' is asked at 3, 6 and 9 months, with the patient and investigator responding with one of the following: stayed the same, improved, got worse.

b. *Percentage repigmentation:* recorded by investigators at 3, 6 and 9 months clinic visits for each of the assessed patches. In addition, an independent dermatologist will provide blinded assessment based on photos of the target patch at baseline and 9 months.

c. Pattern of repigmentation (perifollicular, marginal, diffuse, mixed, hyperpigmentation) will be recorded for descriptive purposes.

d. *Quality of life:* assessed at three time points: baseline, end of treatment (9 months) and end of follow-up (21 months). VitiQOL (vitiligo-specific), Skindex 29 (dermatology-specific) and EQ-5D-5L (generic health-related quality of life) are completed by adults aged 18 years and above.[24–28] All children up to and including 17 years of age at randomisation will complete the CHU 9D (children's health-related quality of life), and children aged 11 years and above at randomisation will complete the EQ-5D-5L (generic health-related quality of life).[28–31]

e. *Time burden of treatment:* participant-reported treatment burden at 3, 6 and 9 months based on average duration and number of treatment sessions and adherence with the treatment schedule. To be presented for light therapy and topical corticosteroid therapy separately.

### Secondary outcome during treatment and follow-up phase

a. *Participant-reported treatment success, analysed by body region:* using the VNS, assessed for each patch at 3, 6 and 9 months, and again in follow-up questionnaires at 12, 15, 18 and 21 months, to assess long-term patient-reported noticeability for each body region.

b. *Resource use to inform within-trial cost-effectiveness analysis from an NHS perspective (primary) and a*

*family perspective (secondary).* Participant self-report of healthcare appointments (number, which professional, and if related to vitiligo), prescriptions and personal expenses, in relation to vitiligo over the course of the treatment and follow-up phases.

### Secondary outcome during follow-up phase only

*Maintenance of treatment response:* assessed by participants for each patch of vitiligo in each of the three body regions during the long-term follow-up period (12, 15, 18 and 21 months questionnaires). The question asked is "Thinking now about since you stopped using the study treatments, has the vitiligo on your [*Head and Neck, Hands and Feet, Rest of Body*] patch: stayed the same, improved, got worse?" Since this response is a subjective personal assessment of how the treatment response is perceived, participants are not shown their end of treatment photograph when making this judgement.

### Safety outcomes

The safety end points are proportion of adverse device effects and adverse reactions to the topical corticosteroid and NB-UVB during the treatment phase.

Adverse events are recorded in the trial database, and monitored by the trial coordinating centre and relevant oversight committees. Erythema (redness) of grade 1 or 2 is not considered an adverse event as this is an expected treatment response from use of NB-UVB. All serious adverse events (SAEs) are reported directly to the trial coordinating centre and assessed for seriousness, expectedness and causality by the Chief Investigator, or delegated medical monitor. SAEs are recorded and reported to the Medicines Health Regulatory Authority (MHRA) and Research Ethics Committee (REC) as part of the annual reports. Serious unsuspected serious adverse reaction (SUSARs) will be reported within the statutory timeframes to the MHRA and REC.

### Data, monitoring and analysis

#### Data collection and management

Data collection and skin assessments are undertaken by trained staff at the recruiting hospitals. Participants who do not attend clinic visits at 3 or 6 months continue to be invited to subsequent follow-up visits unless they decline further participation in the trial. Every effort is made to ensure the 9-month follow-up visit takes place, and if a face-to-face visit cannot be arranged, data are collected via telephone or email if possible. Data captured at clinic visits are entered by site staff into the web-based trial database.

The year-long follow-up after treatment is by self-completed questionnaires, either via an email link to the trial online system or by post, sent with a stamped addressed envelope. Reminders are sent (via email or post) if the questionnaire remains uncompleted after 2 weeks, and again after 3 weeks. The trial coordinating centre also then chases up outstanding questionnaires after 3 weeks via telephone.

Detailed data management processes and central and on-site monitoring procedures are documented in a data management plan. All sites are monitored at least once after the first five randomisations from that site, and triggered monitoring visits are undertaken if any concerns about the site are raised. Database validation checks are conducted centrally including checks for missing data, out of range values, illogical entries and invalid responses. Data entered by sites into the trial database are subject to monitoring and review by coordinating centre staff, and data queries are raised as necessary. All receive at least one site monitoring visit during the trial.

### Trial management

Management of the trial is the responsibility of the Trial Management Group, which meets regularly, usually monthly, throughout the trial. Data collection, adherence and retention rates are monitored by this group. Trial oversight is by the independent Trial Steering Committee. Unblinded trial data are monitored in confidence by the independent Data Monitoring Committee, which reports to the Trial Steering Committee. Both oversight committees meet at least annually. The composition of both oversight committees and their charters can be found on the trial website.

### Sample size

Our choice of minimum clinically important difference between the groups has been informed by a survey of the clinical membership of the UK Dermatology Clinical Trials Network. Assuming that topical corticosteroid alone results in 15% treatment success at 9 months,[12] clinicians were keen to see at least a 20% absolute difference between the groups if they were to justify the cost and complexity of introducing home-based NB-UVB as a potential new service in clinical practice (either on its own or in combination with topical corticosteroid).

The following sample size estimate formed the basis of the funding application and protocol at the start of the study: based on the assumption that 15% of participants allocated to receive topical corticosteroid alone achieve treatment success,[12] 372 participants are required to detect an absolute difference of 20%, with 2.5% two-sided alpha and 90% power. Two comparisons are of coprimary interest, light therapy and light therapy plus topical corticosteroid each versus topical corticosteroid alone. Allowing for up to 15% non-collection of primary outcome data at 9 months, the target sample size was 440 participants.

As data were limited to inform the sample size calculation for this trial, a planned sample size review by the Data Monitoring Committee was scheduled after 18 months of recruitment. This review was conducted in December 2016 and resulted in a recommendation to increase the sample size by a further 76 participants in order to maintain 90% power to detect a risk difference of 20% between the topical corticosteroid arm and the other two arms. The final sample size estimate is therefore 516

participants. This recommendation was approved by the Trial Steering Committee and the funders.

## Statistical methods

The primary approach to between-group comparisons will be to analyse participants according to the group to which they were allocated, regardless of treatment received, levels of adherence and with multiple imputation of missing outcome data.

Baseline data will be presented descriptively, and presented by intervention group.

For the primary outcome, number and percentage of participants achieving 'treatment success' will be reported for each treatment group at 9 months from randomisation. Randomised groups will be compared using a generalised linear model for binary outcome adjusted by centre, body site of vitiligo and age. The primary effectiveness parameter for the two coprimary comparisons of topical corticosteroid versus NB-UVB light, and topical corticosteroid versus NB-UVB light plus topical corticosteroid, will be the risk difference in the proportion of participants achieving treatment success at 9 months, along with 95% CI and exact P value. We will also report relative effects using risk ratios. Sensitivity analyses will be conducted to (i) further adjust for any variables with marked imbalance at baseline, (ii) repeat primary analysis based on participants whose primary outcome is available at 9 months and (iii) investigate the effects of treatment adherence. Planned subgroup analyses are (i) children versus adults and (ii) by body region of the target vitiligo patch. Further secondary analyses will be defined in the statistical analysis plan prior to locking the trial database. These analyses will be conducted by inclusion of appropriate interaction terms in the regression model, and will be considered as exploratory.

Analyses investigating other follow-up times, treatment success of patches on other body sites and other secondary outcomes will be analysed by a similar approach, using appropriate regression modelling depending on outcome type.

All analyses will be specified in a statistical analysis plan to be finalised and approved by the Data Monitoring Committee and Trial Steering Committee prior to locking the database.

## Interim analyses

There are no planned interim between-group analyses.

## Health economic evaluation

An incremental cost analysis will be conducted from an NHS perspective capturing the intervention resource use and other health resource use throughout the treatment and follow-up period (21 months in total). In addition to an NHS perspective, but presented separately, an estimate of out-of-pocket and time costs of treatment for participants and parents/guardians will be recorded. Intervention and wider healthcare costs will be estimated during the trial through participant dairies and case report forms (CRFs). Resource use will be valued using published unit costs or participant-reported estimates for a common price year. Both costs and benefits will be discounted using recommended rates.[32]

Both an incremental cost-effectiveness analysis using the trial primary outcome to estimate the incremental cost per successful treatment and cost-utility analysis estimating incremental cost per quality-adjusted life years (QALY) will be conducted. Utility will be measured at baseline, 9 months and 21 months using the EQ-5D-5L[26–28] (for adults and children aged 11 years or over) and CHU-9D[28–31] (for children aged 5–17 years) questionnaires, which will be used to estimate the QALY over the study period using linear interpolation and area under the curve analysis with and without baseline adjustment.[33] In the base case, separate cost utility analyses will be presented for those aged 18 years and over using the EQ-5D-5L to estimate QALYs and for those aged under 18 years using the CHU-9D. Where appropriate (ie, in the absence of either intervention dominating), the incremental cost-effectiveness of NB-UVB alone compared with TCS alone and separately of NB-UVB plus TCS compared with TCS alone will be estimated by dividing the difference in costs by the difference in QALYs. Decision uncertainty will be presented via Cost-Effectiveness Acceptability Curves based on non-parametric bootstrapping of cost and effect pairs.[34 35] This will provide robust trial evidence to inform decision makers about the likely cost-effectiveness of interventions for vitiligo in particular about whether NB-UVB light is more cost-effective than topical corticosteroid alone and about whether combination treatment offers greater value for money than TCS treatment alone.

## Patient and public involvement

People with vitiligo have helped to prioritise this research question through a James Lind Alliance Priority Setting Partnership, which highlighted this as a priority topic.[16] A patient representative was a member of the Trial Development Group, a coapplicant on the funding award and has contributed to the design and conduct of the trial throughout (including the external pilot RCT,[15] and development of the training video (found at www.vitiligo-study.org.uk).

Our choice of outcome measures was informed by feedback from people with vitiligo that suggested a disparity between what has previously been measured in vitiligo trials and what patients feel is important.[21] They also contributed to the choice of outcome measures, through involvement in development of the core outcome set for vitiligo,[22] and in development and validation of the primary outcome scale (VNS).[23 36]

Throughout the trial, people with vitiligo have contributed to the development of training materials and have commented on the suitability of participant-facing materials and questionnaires. They are continuing to contribute to awareness and engagement activities. The trial is supported by the UK Vitiligo Society and a patient

**Table 3** Summary of protocol amendments that impacted on trial design

| Protocol | Date | Summary of changes |
|---|---|---|
| 2.0 | 11 March 2015 | Added details of the MRC systematic techniques for assisting recruitment to trials (START) substudy |
| 3.0 | 30 Sepember 2015 | Clarified inclusion and exclusion criteria; added more details about training participants to use trial treatments; procedures clarified for digital images outcome analyses; changes to adverse events (AE) handling for erythema (grade 1 and 2 are not AE, but expected reactions) and amendment of prespecified subgroup analysis to remove a comparison of active and inactive patches (as by definition all target patches will be active), and add a subgroup analysis evaluating response of target patch by region of the body. |
| 4.0 | 03 March 2017 | Added details of the nested process evaluation; updates to the safety handling section; introduction of an online automated the blind brake procedure; change to sample size following sample size review by the Data Monitoring Committee (DMC). |
| 5.0 | 18 January 2018 | Due to trial timelines some participants will not receive the full 12 month follow-up but will receive quality of life questionnaires and study feedback questions; Updates to statistical analyses section to reflect the statistical analysis plan; addition of output testing of NV-UVB devices after end of treatment phase. |

representative provides trial oversight as a member of the Trial Steering Committee. A panel consisting of three people with vitiligo will assess the digital images to provide blinded outcome assessment of the primary outcome.

### Ethics and dissemination

#### Protocol amendments

The methods described in this protocol reflect the current protocol (V.5.0 dated 18 January 2018). A summary of protocol amendments that took place after start of recruitment are summarised in table 3.

#### Consent

Age-appropriate participant information sheets are provided for all trial participants, and ample opportunity is given to discuss the study with the study team before agreeing to take part. All participants will provide written informed consent (or assent in the case of children under 16) for participation, retention and use of the trial data by members of the research group (use of the trial photographs for further research or trial reporting is optional).

#### Confidentiality

Trial data and individual participants' medical information obtained as a result of this study is considered confidential. Participant confidentiality is ensured by using identification code numbers to correspond to treatment data in the computer files and on trial photographs.

#### Post-trial care

After completing the trial, participants will continue to receive normal vitiligo care in accordance with local practice. A summary of the trial results will be sent to participants if they have given consent for this.

#### Dissemination policy

Results will be reported in full through the National Institute for Health Research Journal series (open access), as well as through peer-reviewed journals, conferences,

participant newsletters, patient focused charities and websites.

### DISCUSSION

The HI-Light Vitiligo Trial is an appropriately powered, pragmatic, RCT that addresses a topic that has been identified as important by people with vitiligo, healthcare professionals who treat them and by the NIHR. It is unique in that it includes both children and adults with early and limited vitiligo, places particular emphasis on patient-reported outcomes and has been delivered with the support of a professional Clinical Trials Unit.

We support the development and use of core outcome sets in clinical trials and have included outcomes that are currently recommended as part of the core outcome set for vitiligo.[22] However, outcomes research is rapidly evolving. Since starting the trial, further guidance has emerged[37] suggesting that 80%, rather than 75% repigmentation, should be used to represent a meaningful treatment response, and that maintenance of repigmentation should be evaluated 6 months after stopping treatment. We will consider this updated guidance when finalising our statistical analysis plan and final write-up, to ensure that data are presented in a way that will facilitate comparison with other trial datasets in the future.

Alongside the trial, a mixed methods process evaluation study is being conducted to explore barriers and facilitators to use of the trial interventions in normal care. The views of trial participants, healthcare professions and commissioners on the use of the trial interventions will be sought. A process evaluation can help to explain trial findings, explore how intervention delivery within the trial may differ from 'real-world' delivery and identify issues important to the transferability of an effective intervention outside the trial.[38] This process evaluation is particularly relevant for a study such as HI-Light, where hand-held NB-UVB devices represent a new way

of delivering care with complexity around training and treatment adherence.

Several resources have been developed during the set-up of this trial that may be helpful for other researchers wishing to conduct trials of home light therapy, or for health providers wishing to implement the study findings. A training video demonstrating how to use home NB-UVB is freely available, along with training materials for conducting MED tests and copies of study documents and treatment diaries (www.vitiligostudy.org.uk). The trial is also contributing to a nested methodological study (START) looking at the value of interactive websites to facilitate recruitment and retention in trials (http://research.bmh.manchester.ac.uk/mrcstart). Results of the process evaluation and START study will be reported separately.

## Trial status

The HI-Light Vitiligo Trial is ongoing. The first participant was randomised into the trial in July 2015 and recruitment is anticipated to be completed by Quarter 4 2017.

## Author affiliations

[1]Nottingham Clinical Trials Unit, University of Nottingham, Queen's Medical Centre, Nottingham, UK
[2]Centre of Evidence Based Dermatology, University of Nottingham, Nottingham, UK
[3]Department of Paediatric Dermatology, Nottingham Children's Hospital, Nottingham University Hospitals NHS Trust, Nottingham, UK
[4]Department of Medical Physics and Clinical Engineering, Nottingham University Hospitals NHS Trust, Nottingham, UK
[5]Nottingham Clinical Trials Unit, University of Nottingham, Nottingham Health Science Partners, Queen's Medical Centre, Nottingham, UK
[6]Norwich Medical School, University of East Anglia, Norwich, UK
[7]Primary Care and Population Sciences, University of Southampton, Southampton, UK

**Acknowledgements** The trial is sponsored by the University of Nottingham and supported by the NIHR Clinical Research Network, Nottingham Clinical Trials Unit and the UK Dermatology Clinical Trials Network. The authors are grateful for all those who have contributed to the development and conduct of the study and to the people with vitiligo who are taking part. Dermfix Limited has worked with the trial team to develop the dummy devices and the organisation of device supply (at a discounted rate): the authors thank them for their valuable support. Mawdsleys services in distributing the trial treatments to participants contributes towards the smooth running of the trial. Involvementin the MRC START study was coordinated by Shelley Dowey and the process evaluation by Joanne Chalmers. The authors would like to thank the recruiting centres: Cannock Chase Hospital and New Cross Hospital, The Royal Wolverhampton NHS Trust (PIs: Seau Tak Cheung, and Walter Machado); St Luke's Hospital, Bradford Teaching Hospitals NHS Foundation Trust (PI: Andrew Wright); University Hospital Wales, Cardiff and Vale University Health Board (PI: John Ingram); The Norfolk and Norwich University Hospital, Norfolk and Norwich University Hospitals NHS Foundation Trust (PI: Nick Levell); Solihull Hospital, Heart of England NHS Foundation Trust (PI: Jon Goulding); West Glasgow Ambulatory Care Hospital, NHS Greater Glasgow and Clyde (PI: Areti Makrygeorgou); Whipps Cross Hospital and The Royal London Hospital, Barts Health NHS Trust (PI: Anthony Bewley); Birmingham Children's Hospital, Birmingham Children's Hospital NHS Foundation Trust (PI: Malobi Ogboli); York Hospital, York Teaching Hospital NHS Foundation Trust (PI: Julia Stainforth); Royal Derby Hospital and the London Road Community Hospital, Derby Teaching Hospitals NHS Foundation Trust (PI: Jonathan Batchelor); Queens Medical Centre, Nottingham University Hospitals NHS Trust (PI: ane Ravenscroft); Chelsea and Westminster Hospital, Chelsea and Westminster Hospital NHS Foundation Trust (PI: Bisola Laguda); University Hospital of North Durham; County Durham and Darlington NHS Foundation Trust (PI: Shayamal Wahie); The James Cook University Hospital, South Tees Hospitals NHS Foundation Trust (PI: Rob Ellis); Birmingham City Hospital, Sandwell and West Birmingham Hospitals NHS Trust (PI: Amirtha Vani Rajasekaran).

**Contributors** JB is the Chief Investigator and KST is lead investigator on the grant. HCW, VE, JCR, DW, AR, THS, MS, PA, JRC, LD, AAM, EJM and RHH contributed to the design of the study. GM and RHH are responsible for managing the trial from Nottingham CTU, with the help of JW. AAM, WT and TH are responsible for the statistical analysis plan. THS is the lead for the health economic component. RHH wrote the initial manuscript with KST and JB. All authors reviewed and approved the manuscript.

**Funding** This protocol paper summarises independent research funded by the National Institute for Health Research (NIHR) under its Health Technology Assessment Programme (project number 12/24/02). The pilot RCT was funded by the National Institute for Health Research (NIHR) under its programme grants for applied research (RP-PG-0407-10177). The HI-Light Vitiligo Trial purchased the devices from Dermfix at a discounted rate. Dermfix assisted in sourcing dummy devices for the trial and in arranging logistics of supply, but have had no input into the design or conduct of the trial. The UK Dermatology Clinical Trials Network receive infrastructure funding from the British Association of Dermatologists.

**Disclaimer** The views and opinions expressed therein are those of the authors and do not necessarily reflect those of the Health Technology Assessment Programme, NIHR, NHS or the Department of Health. The views expressed are those of the author(s) and not necessarily those of the NHS, the NIHR or the Department of Health.

**Competing interests** HCW is Director of the NIHR Health Technology Assessment Programme. THS holds a Career Development Fellowship (NIHR-2014-07-006) supported by the National Institute for Health Research.

**Patient consent** Obtained.

**Ethics approval** Prior to the start of recruitment, appropriate approvals were granted by MHRA (EudraCT No. 2014-003473-42), NHS Health Research Authority, NRES Committee East Midlands—Derby (REC reference 14/EM/1173) and NHS Research & Development departments for each participating site.

**Provenance and peer review** Not commissioned; externally peer reviewed.

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
