## [Reviewer comments · BMJ Open]

ARTICLE DETAILS

TITLE (PROVISIONAL)	Home Interventions and Light therapy for the treatment of vitiligo (HI-Light Vitiligo Trial): study protocol for a randomised controlled trial
AUTHORS	Haines, Rachel; Thomas, Kim; Montgomery, Alan; Ravenscroft, Jane; Akram, Perways; Chalmers, Joanne; Whitham, Diane; Duley, Lelia; Eleftheriadou, Viktoria; Meakin, Garry; Mitchell, Eleanor; White, Jennifer; Rogers, Andy; Sach, Tracey; Santer, Miriam; Tan, Wei; Hepburn, Trish; Williams, HC; Batchelor, Jonathan

VERSION 1 – REVIEW

REVIEWER	Amit Pandya University of Texas Southwestern Medical Center, Dallas, Texas, U.S.A.
REVIEW RETURNED	02-Aug-2017

GENERAL COMMENTS	Well-designed, large study to determine the efficacy of home phototherapy and a potent topical corticosteroid in the treatment of vitiligo from a world leader in evidence based dermatology. My only concern is the possibility of a large number of patients choosing a hand lesion as their primary target lesion. Not much improvement is likely with long-standing hand lesions and if a majority of patients choose hand lesions the results will not be very good.
---

REVIEWER	Furen Zhang Shandong Provincial Institute of Dermatology and Venereology
REVIEW RETURNED	11-Aug-2017

GENERAL COMMENTS	The manuscript here described a well-designed randomized controlled trial to assess the comparative effectiveness of potent topical corticosteroid, home-based hand-held narrowband ultraviolet B-light (NB-UVB) or combination of the two, for the management of vitiligo. Most of methodological designs regarding objective, inclusion and exclusion criteria, statistics, intervention and the device are sound. Although the implementation of the RCT may be very difficult, the outcome of the research will greatly benefit the treatment strategy of vitiligo worldwide. As the treatment phase is quite long, my only concern is how does the doctor make sure the patients appropriately use the device on schedule?
---

VERSION 1 – AUTHOR RESPONSE

Reviewer: 1

Reviewer Name: Amit Pandya

Institution and Country: University of Texas Southwestern Medical Center, Dallas, Texas, U.S.A.

Please state any competing interests or state 'None declared': None declared

Please leave your comments for the authors below

Comment: Well-designed, large study to determine the efficacy of home phototherapy and a potent topical corticosteroid in the treatment of vitiligo from a world leader in evidence based dermatology.

We thank you very much for your encouraging comments.

My only concern is the possibility of a large number of patients choosing a hand lesion as their primary target lesion. Not much improvement is likely with long-standing hand lesions and if a majority of patients choose hand lesions the results will not be very good.

##As the trial places emphasis on treatment outcomes which are meaningful for patients with vitiligo, we wanted patient choice to be a deciding factor in which patches to treat as a part of the trial. Secondly, all target lesions need to have been active in the last year (i.e. none will be "long-standing" lesions), so all of these lesions should be more susceptible to any potential improvements, if the treatments are effective. However, it was highlighted in the site training materials that investigators should make participants aware that patches on the hands and feet may be less likely to respond to treatment. Further details to address this comment have been added on page 12 of the manuscript.

At present, only 31% of recruited participants have chosen a target patch in the hands and feet region, and further secondary subgroup analyses by body region (on patches other than the target patch) will also be performed.

Reviewer: 2

Reviewer Name: Furen Zhang

Institution and Country: Shandong Provincial Institute of Dermatology and Venereology

Please state any competing interests or state 'None declared': No

Please leave your comments for the authors below

Comment: The manuscript here described a well-designed randomized controlled trial to assess the comparative effectiveness of potent topical corticosteroid, home-based hand-held narrowband ultraviolet B-light (NB-UVB) or combination of the two, for the management of vitiligo. Most of methodological designs regarding objective, inclusion and exclusion criteria, statistics, intervention and the device are sound.

Although the implementation of the RCT may be very difficult, the outcome of the research will greatly benefit the treatment strategy of vitiligo worldwide.

We are pleased to hear this and hope that our nested process evaluation will help to inform development of implementation strategies depending on the results of the trial.

Comment: As the treatment phase is quite long, my only concern is how does the doctor make sure the patients appropriately use the device on schedule?

##As this is a pragmatic RCT we are capturing adherence as it might happen in normal practice. Adherence is recorded by participants in treatment diaries which are checked at each clinic visit; clinic visits were chosen at 3 month intervals to reflect what might happen in normal practice (at least in the UK). Trial participants receive a 2-week post randomisation telephone call to check their understanding of the use of both treatments as well. We will monitor adherence closely in the trial and explore the impact of poor adherence on treatment response.

To provide further clarity about appropriate usage and adherence monitoring, we have added additional text on pages 12 and 13 of the manuscript.

VERSION 2 – REVIEW

REVIEWER	Amit Pandya University of Texas Southwestern Medical Center
REVIEW RETURNED	02-Oct-2017
GENERAL COMMENTS	Well designed and important study